# Damage Behavior with Atomic Force Microscopy on Anti-Bacterial Nanostructure Arrays

**DOI:** 10.3390/nano14030253

**Published:** 2024-01-24

**Authors:** Jonathan Wood, Richard Bright, Dennis Palms, Dan Barker, Krasimir Vasilev

**Affiliations:** 1Future Industries Institute, University of South Australia, Mawson Lakes, SA 5095, Australia; jonathan.wood@mymail.unisa.edu.au; 2College of Medicine and Public Health, Flinders University, Bedford Park, SA 5042, Australia; dennis.palms@flinders.edu.au; 3Corin Australia, Sydney, NSW 2153, Australia; dbarker@anisop.com.au

**Keywords:** atomic force microscopy, AFM, LFM, deformation, hydrothermally etched, nanostructures

## Abstract

The atomic force microscope is a versatile tool for assessing the topography, friction, and roughness of a broad spectrum of surfaces, encompassing anti-bacterial nanostructure arrays. Measuring and comparing all these values with one instrument allows clear comparisons of many nanomechanical reactions and anomalies. Increasing nano-Newton-level forces through the cantilever tip allows for the testing and measuring of failure points, damage behavior, and functionality under unfavorable conditions. Subjecting a grade 5 titanium alloy to hydrothermally etched nanostructures while applying elevated cantilever tip forces resulted in the observation of irreversible damage through atomic force microscopy. Despite the damage, a rough and non-uniform morphology remained that may still allow it to perform in its intended application as an anti-bacterial implant surface. Utilizing an atomic force microscope enables the evaluation of these surfaces before their biomedical application.

## 1. Introduction

Utilizing a single instrument, namely atomic force microscopy (AFM), enables the simultaneous acquisition of nanomechanical and topographical data from nanostructured surfaces [1]. Using the same instrument for all surface analysis is not only efficient, but data are also calibrated parallel to each other in a comparable format and in the same environment using the same measuring probe. Hydrothermally etched nanostructures of Ti6Al4V generate an irregular surface, posing challenges in achieving a precise range of surface interaction values necessary for anti-bacterial biomedical applications [2,3,4]. This is shown in the behavior of a micron-scale spherical cantilever tip contacting several Ti6Al4V nanostructures at a time [5]. Damage to an individual or a small group of these nanostructures is expected to vary in behavior as they undergo elastic and plastic deformation under a certain load [6]. Forces strong enough to deform and permanently damage the Ti6Al4V nanostructured surface, intended primarily for end use as an anti-bacterial implant surface, are dependent on the morphology of the individual and groups of nanostructures. The variations in the patterning, height, width, angle, spacing, etc., of the nanostructures will alter the minimum force to deform them. A high enough force to plastically deform a surface would be expected to alter the existing patterning, and in turn, affect the morphologically dependent functioning of the nanostructured surface [7,8] It is anticipated that the damage to the nanostructure array of Ti6Al4V will lead to a certain level of modification in the efficacy of bacterial eradication. Assessing fundamental interactions through the AFM reveals the advantages and disadvantages that guide ongoing design enhancements. A comprehensive understanding of how these structures respond, along with an awareness of the error limits impacting their functionality, is crucial for optimizing performance and advancing scientific knowledge on the utilization of intricate surface structures.

## 2. Materials and Methods

### 2.1. Preparation of Hydrothermally Etched Samples

Surface nanostructures were created by subjecting grade 5 titanium alloy (Ti6Al4V) samples to a hydrothermal etching process. The Ti6Al4V discs, sourced from Hamagawa Industrial (M) SDN BHD Sungai Petani, Kedah, Malaysia, were polished and measured 10 mm in diameter and 3 mm in height, with a surface area of 0.78 cm^2^. The initial surface roughness (Ra) of these discs was 0.5 µm. Spike-like nanostructured surfaces were generated by subjecting the discs to hydrothermal etching at 150 °C, utilizing a stainless-steel reactor supplied by Parr Instrument Company (Moline, IL, USA), with 1 M KOH as the etchant. After etching, the reactors were cooled, and the samples were thoroughly cleaned using ultrapure water. Subsequently, the Ti6Al4V discs were dried and underwent heat treatment inside an oven. Following this, they were allowed to cool overnight. The as-received titanium alloy discs were retained as controls (AR-Ti), while the hydrothermally etched discs, treated with 1 M KOH aqueous etching solutions (AMK), were further used for the proceeding analysis [2].

### 2.2. Nanostructure Morphology by Scanning Electron Microscopy (SEM)

The morphology and distribution of nanostructures on the surface of the titanium substrate were examined using a Zeiss Merlin FEG-SEM (Zeiss, Jena, Germany). The SEM was equipped with a secondary electron (SE) detector and operated at 2 KV, allowing for magnification ranging from 5 to 50,000 times. During the imaging of nanostructures, the stage was tilted at 45 degrees, while it was kept orthogonal when analyzing the density of individual nanostructures and imaging bacteria on the surfaces. The height of the nanostructures was determined by measuring the linear distance between the basal plane and the highest point of each spike, and the diameter of the nanostructures was measured at the midpoint, in parallel orientation with the basal plane. The spike height and diameter at mid-height were presented as the mean ± standard deviation, with a sample size of *n* = 5. The dimensions of the nanostructures were assessed using ImageJ software version 1.53f51 (NIH, Bethesda, MD, USA).

### 2.3. Cantilevers

NT-MDT brand NSG30 and NSG03 conical tipped SiN cantilevers (NT-MDT, Moscow, Russia) and a high spring constant 5 µm diameter in SiO_2_ spherical tip cantilevers were used. NT-MDT NSG03 has a spring constant generally in the 1–5 N/m range, and an NSG30 cantilever has a spring constant usually in the 22–100 N/m range. These N-type conical cantilevers possess a half-side angle of 18° and a tip apex radius < 10 nm. The 5 µm diameter spherical tip cantilever has a spring constant between 20 and 70 N/m. Tip wear over scanning changes the morphology of the tip apex. The values provided are for unused cantilevers and are quoted from the manufacturer.

### 2.4. Atomic Force Microscopy (AFM) Analysis

All measurements were performed using a JPK NanoWizard III AFM (Bruker, MA, USA), with results being processed in JPK NanoWizard data processing software, Gwyddion freeware version 2.65 (http://gwyddion.net/download.php, (accessed on 18 January 2024) Czech Metrology Institute, Brno, Czech Republic), ImageJ Version 1.53t (NIH, Bethesda, MD, USA), and Office Excel 365 ProPlus. Topographical measurements were performed in both tapping and contact modes with raster scanning of sample areas beginning from the bottom left corner and ending at the top right corner with a vertical slow scan direction. Topographical measurements using the AFM can be performed either in tapping or contact mode. The tapping mode oscillates the AFM cantilever, making intermediate contact with the sample surface. Contact mode topography has the tip in constant contact with the surface. This mode produces greater wear than the tapping mode but allows coinciding friction measurements to be recorded as Lateral Force Microscopy (LFM) data [9]. Roughness and characterization of forces were detailed through topography, with contact mode topography primarily used for the measurement of friction coefficient and friction force in LFM [10,11]. AFM scans were colored, processed, and measured for roughness values, feature heights, image cropping, and 3D representations in Gwyddion software. Friction data were imported into Microsoft Excel 365 for friction coefficient and friction force calculations, averages, and 2D plots.

### 2.5. Statistical Analysis

Analysis and data visualization were carried out using Gwyddion software, version 2.65, and Microsoft Office Excel. All experiments were conducted in triplicate unless otherwise specified. Data are presented as mean and standard deviation (SD).

## 3. Results and Discussion

### 3.1. Nanotopography Analysis by SEM

Emerging as seemingly haphazard arrangements, spaced apart to a greater degree due to collapse and aggregation, these structures take on a tower-like form, joined at their uppermost layers. Occasionally, a taller structure extends upward, like an antenna, from the cluster. The differences in patterning and morphology of the AMK nanostructures are shown in the Scanning Electron Microscopy images in Figure 1. The SEM images were utilized to measure the dimension and distribution of the nanospikes (Figure 1A and Figure 2A). The height of the spikes, mid-height diameter, and spacing between joined ‘peaks’ of the clustered nanostructures were 340 ± 174 nm, 83 ± 32 nm, and 544 ± 150 nm, respectively (*n* = 20). Additionally, the density of the AMK nanostructure surface was 8 ± 2 spikes/µm^2^. Comparative AFM images were obtained to contrast with the SEM findings (depicted in Figure 1C,D). The Root Mean Square (RMS) and Roughness Average (Ra) were calculated based on AFM images of the AR-Ti and AMK surfaces (Figure 1E). The Root Mean Square (RMS) and Roughness Average (Ra) for the AR-Ti were 4.1 ± 0.8 nm and 3.2 ± 0.7 nm, respectively. As anticipated, the values for the AMK surface exhibited a significant increase, reaching 73.0 ± 6.0 nm for RMS and 58.2 ± 4.8 nm for the Ra value (*n* = 5).

### 3.2. Nanostructure Damage in Tapping Mode

Continuous scanning over a sample wears and damages both the tip and/or the sample surface. The rate of damage of each solid material is dependent on the geometry, composition, and nanomechanical factors, such as the elastic–plastic ratio. AFM cantilever tips are expected to be worn at the tip apex due to their small surface area. An NSG30 model conical cantilever with a <10 nm tip radius wears noticeably in contact mode after a small number of scans, but this wear is heavily reduced in tapping mode as there is less contact over the scan area and minimal lateral movement across the surface [12]. Wear and damage in tapping mode are heavily dependent on the SetPoint force, which is the applied contact force per tap [13,14]. Standard tapping mode fixes the oscillation amplitude of the cantilever during scanning. Monitored by the instrument’s feedback loop, the amplitude can alter changes in topography and material composition. Quickly responding, the feedback loop adjusts the cantilever sample distance to maintain the pre-set amplitude value. Fixing the cantilever’s amplitude results in changes in the applied contact force between the sample tip, particularly due to topographical variations. A varied tip contact force is applied across the scan with each tap, although the force values are expected to be within a narrow range [15]. This can result in uneven sample surface damage. Nanostructures have been experimentally recorded as being damaged with the nano-Newton force range applied by the AFM. The high level of topographical variation over a nanostructure array creates a high variation in cantilever amplitude, thus creating a variation in the applied force by the cantilever. Damage can occur over only a few scans. Figure 2 is a progression of 5 × 5 µm scans over AMK nanostructures performed in air. After each scan, the sample was moved 1 µm to the right on the x-axis. This meant that the left 1 µm section was scanned once. The next 1 µm section was scanned twice, the next section was scanned three times, and so on. A map is progressively built where damage to the nanostructure surface graduates after each scan. Clear damage can be seen in the 3D representations, in Figure 2F, where the top level of nanostructure peaks is severely reduced. Graduating in 1 µm steps, the reduction in the upper nanostructure peaks is clear on the right-hand side of the image. Figure 2G zooms into a marked region where the nanostructure peaks have been damaged. Maximum nanostructure heights in this area were around 100 nm, compared to around 350 nm measured at the far lower right of Figure 2D. Flatter, wider, and duller nanostructure patterning is seen across this region of the sample. The area has been flattened, indicating that the nanostructures have been pushed down plastically and are generally lying flat [16]. Collapsed nanostructures are around 25% wider. The SetPoint for these scans was left at the instrument-derived setting of 33.6 nm and a drive amplitude of 0.06 V for a stiff conical-tipped NSG30 cantilever with a spring constant of 28 N/m.

### 3.3. Tapping Mode Area Indentation

Scanning in tapping mode over the same region and then expanding the scan area from 5 × 5 to 20 × 20 µm shows the four 5 × 5 µm scan areas previously performed in Figure 3. In only a small number of scan passes, an indentation was created from the tip. By measuring an 85 nm lowering of the nanostructures in the z-axis in Figure 3C, there was mostly a plastic deformation of the nanostructures, as there was little recovery in the surface z-axis over continued scanning. The cantilever spring constant was 28 N/m, and the SetPoint distance and Drive amplitude in tapping mode were 33.7 nm and 0.06 V, respectively, which do not relate to a high tip contact force considering the surface material and the complex topography of the surface. The considerably lower roughness in the indented area shown in Figure 3C indicates flattened and compressed nanostructures.

Further analysis of the indentation area in Figure 3 shows the considerable damage caused by the NSG30 cantilever (spring constant 27.94 N/m) in tapping mode scanning. Figure 4A crops a region of the 20 × 20 µm scan area to relate the previous 5 × 5 µm scan area against a newly scanned region, as shown in Figure 4B. There is a distinct change in the nanostructure array, as shown at the point marked by a white arrow in Figure 4A. A large region of z-axis depression and patchiness of the nanostructures is shown on the right side of the image. Figure 4C,D maps two 1.5 × 1.5 µm areas to gather roughness data of heavily damaged to low damaged nanostructures shown in the table of Figure 4E. A considerable reduction in RMS and Ra roughness between these areas measures 11 ± 4 nm and almost half the surface area, which relates to nanostructure flattening at a certain minimum z-axis value, creating a slightly more uniform surface height. Figure 4F,G measure the z-axis drop over the previously scanned area at a 200 ± 65 nm difference. This is significant as the nanostructures average around 500–700 nm, deriving a decrease in almost a third of their overall height from the base. Figure 4H measures two nanostructures in the previously scanned region and one nanostructure that appeared to have been negligibly reduced in height measuring 200 ± 34 nm. Inside the 5 × 5 µm previously scanned area, a contrast can be seen between the top and bottom half of the area. The top half does not appear as damaged as the bottom half. The reasons for this are unknown, but the most likely cause is due to a shift in feedback from the change in measured surface height. This change in height has caused a change in the drive amplitude and thus a change in the measured height of the surface and a change in the force applied by the tip during tapping [17]. Roughness values were taken at consistent areas from the upper and lower regions. RMS roughness in the higher region was at 48 ± 15 nm to 30 ± 12 nm for the lower. Ra contrast was at 35 ± 10 to 21 ± 5 nm, showing a considerable change. 

Increasing the scan area over the same region at 50 × 50 µm, the indentation of the nanostructures by the conical tip is further confirmed with indentation and compression occurring in the z-axis in the 5 × 5 and 20 × 20 µm scan areas. The plot in Figure 5C shows nanostructure compression from 10 to 40 nanometers, with a reduced range of tip penetration between nanostructures indicating flattening. For a uniform flat solid surface, the initial spherical or blunt conical tip contact area at the center point is at a higher force than the graduating outlier contacting areas. This will differ for a nanostructure array where the surface is not uniform and varies largely in the z-axis. Thus, the highest force may not occur at the center of the tip contact, and there may be more than one point of contact. After initial contact with the tip, the pressure gradually increases. At a high SetPoint value, normal force from the tip will, at some point, create a high enough pressure to change from elastic deformation of the nanostructured surface to plastic deformation. A pop-in event can occur around this transition, which is characterized by a sudden displacement at a particular force. This will appear as discontinuous steps in a mapped loading-distance curve. Quite often, these are seen on Berkovich indentation plots, as high-loading forces into the Newtons can be applied [18,19]. On a uniform surface, the first pop-in event is often at the initial dislocation nucleation, which is the elastic–plastic transition. This point is usually close to the theoretical strength of the sample material. The surface material is displaced as the tip, applies more pressure for a uniform solid surface, and is accommodated by the elastic expansion of the surrounding area, creating an elastic zone surrounding the initial plastic zone. The plastic zone is at the center of the contact, as this is the highest region of an applied force by the tip. Increasing the applied force generally results in an increased area of indentation. Increased pressure occurs beneath the tip, radiating out from the center point, which is necessary to allow increased elastic expansion. In other words, the deformation of a uniform surface occurs as the tip negatively displaces the contact area, creating elastic deformation in a secondary area further out from the center of contact. This creates an expansion and compression of the material into the bulk, laterally through the material’s plane, and outside the surface plane as a positive displacement of the surface. If the force is high enough, the plastic zone expands to the free surface and plastically deforms the positively displaced material. This effect is largely determined by the material’s nanomechanical characteristics and composition. Indentation is mostly a non-elastic process, although there are often elastic regions and minor elastic recovery, again depending on material composition [20,21].

### 3.4. Utilizing a Spherical Tip for Nanostructure Compression

Changing the tip geometry in measuring an anti-bacterial nanostructured surface, such as those in Figure 4 and Figure 5, was performed by replacing a conical tip with a large spherical tip to assess nanostructure damage by a larger surface area tip. A larger tip area spreads the applied force over several nanostructures, demonstrating force reactions over an array area rather than focusing on individual structures. A larger tip sample surface contact array measures force reactions as a sample of the array as a whole rather than individual structures, and thus a greater force needs to be applied to re-enact comparable damage, such as a sharper and smaller area tip, akin to a bed-of-nails model. The size and shape of a 5 µm diameter spherical type are aimed at modeling an interacting particle, such as a biological cell. An ideal model of the initial contact is of a single nanostructure at the center of the spherical tip. This is not usually the case over the larger nanostructured surface. Over an increasing applied force by the cantilever tip, many variations of nanostructure tip behavior can occur. Expected modeling behavior would view the force increasing compression of the nanostructure elastically until a pop-in event occurred where the nanostructure would, at first, elastically bend or buckle, and then plastically, depending on the nanostructure’s composition. However, the more likely scenario for a tall and thin nanostructure is lateral bending.

The lateral stiffness of a tall and thin nanostructure would usually be a lot lower than its vertical stiffness. Depending on the spacing between nanostructures, there is a certain degree of freedom where the nanostructure will be pushed both laterally and down into the structure array until the nanostructure contacts neighboring nanostructure/s. As the tip moves down against other nanostructures, under an increasing tip-derived force, these neighboring nanostructures will be bent laterally, not just from the tip but from contact with other already bending nanostructure/s, as shown in Figure 1. This behavior spreads out from the center point of initial contact over the increasing tip nanostructure surface contact zone in all lateral directions. This is considering the tip as a perfect sphere moving at a uniform velocity and force, with the nanostructures at an even height and spacing. However, the nanostructures are not uniform in patterning or morphology, the force and velocity are not uniform, and the tip is not an ideal sphere at the nanoscale. The tip is rough at the nanoscale, as are the nanostructure/s, meaning friction contributes to deformation behavior as the tip slides over the nanostructures [21,22].

Upon the first point of contact, the cantilever could experience minor vertical, lateral, and torsional deviations, coupled with a degree of energy dissipation. This energy dissipation will progressively increase as more force is applied and a greater number of nanostructures are engaged. Deflection and energy dissipation can change the modeling of this action dramatically and will be different for each interaction. Nanostructures grown on the surface are not often uniform in height, spacing, diameter, apex shape, and angle to the surface plane. This means that the initial secondary tip nanostructure contacts may occur at any point of the facing spherical tip and as the tip moves further into the surface. Additionally, the meniscus layer needs to be accounted for. When exposed to air, a solid surface will develop an atmospheric water layer. The layer thickness will depend on relative humidity, surface morphology, and chemistry. Nanostructure spacing past a certain point will not allow a consistent meniscus layer; however, depending on the diameter and morphology of the individual structure, small meniscus layers may be present on each or many structures. This factor will affect initial contact and even alter the lateral bending under applied tip force due to the adhesion of the tip to nanostructures and the nanostructures to each other [23,24].

### 3.5. Half-Sphere Scan Anomaly 

A contact mode topography analysis was performed on the AMK nanostructured samples in ambient air, utilizing a SiO_2_ spherical cantilever tip with a 5 µm diameter and a spring constant of 40.8 N/m. Topography scans are shown in Figure 6, with the scan area starting at 5 × 5 µm and increasing by 1 × 1 µm on each subsequent scan to display any significant changes in the overall nanostructure surface from the previous scans. The SetPoint force, which is the cantilever force applied normally to the surface, started at 320 nN for the 5 × 5 µm scan. A micron-diameter spherical cantilever is used as a geometric model, mimicking a bacterial or mammalian cell. A high spring constant spherical cantilever applying a higher force than a cell would initially contact in a natural scenario. Forces of greater than 10 nN have been shown to induce irreversible damage to the cell membranes of both Gram-negative Pseudomonas aeruginosa [4] and Gram-positive Staphylococcus aureus bacteria [25], as well as mammalian cells, including red blood cells, somatic cells, and phospholipids [26,27]. Due to a larger area of contact, measured in a previous experiment of a 1 × 1 µm contact area at a force of 100 nN on a Ti6Al4V polished control sample, the level of force required to cause damage is expected to be a lot higher. Contact area calculations are based on a flat continuous surface and do not account for the significant reduction in contact area due to an irregular surface. An increase in the cantilever’s SetPoint will push further into the nanostructure array, causing contact with a greater number of nanostructures on an increasing area of the spherical tip. The greater the number of contacting nanostructures, the less pressure is applied to the individual structure. Additionally, the AMK surface consists of many layers of nanostructures. As the tip pushes into the nanostructures, a dual action increase occurs from the lateral increase in the number of contacts in the x-y axis and the z-axis from contacting additional nanostructure layers.

Figure 6A–D show a progressive increase in the resolution of most nanostructures as the SetPoint force increases in 50 nN steps from 320 nN to 470 nN. Figure 6D shows an outline of the 5 µm diameter cantilever tip as a scale reference to the maximum possible area of contact. It is generally expected that the higher the applied force, the greater the area of contact. An applied force that would damage the nanostructures over this area was experimentally shown to be in the µN range. The maximum applied force by the cantilever tip is shown in Figure 6E at 5200 nN (5.2 µN). This magnitude of applied force heavily damaged the nanostructures over the entire scan region. Nanostructures were plastically damaged, as shown in a follow-up scan in Figure 6F at a SetPoint force of 520 nN. In this scan, there is no discernible individual structure resolution and only minor recovery of the nanostructures. The 1 × 1 µm larger scan area in Figure 6G shows the undamaged edge past the last higher SetPoint force scan area. A 155 nm indented region is measured in the z-axis at a point between the previous scan area of 5200 nN and the edge of the following 520 nN scan, as shown by the plot in Figure 6H.

There are several artifacts present in spherical tip scanning of nanostructure surfaces, as shown in Figure 6. These include edge overshoots that appear as narrow hills and valleys at the edge of surface terraces. Occurring when the tip moves over a step edge, they are generated by piezoelectric scanner hysteresis and/or a potential barrier and are seen to a small extent with closely spaced nanostructures. A sudden decline and subsequent rise in elevation may give the impression of a narrow area that appears to be higher than the actual topography. Other scan image effects may also make the edges appear more pronounced as there is a raised edge of the sphere present, instead of the expected drop in the z-axis as the spherical tip gradually moves off the nanostructure. Scanning at a faster velocity can increase this artifact. Edge overshoot appears as the same artifact feature, but it does not increase with an increased scan velocity. Large attractive forces can occur at the step edges due to changes in nanostructure uniformity and chemistry. Anomalies such as these can be complicated by the flexion and movement of the nanostructures under an applied force and can affect resolution as a scale of deformation. Usually, these effects refer to soft samples, such as polymers and biological materials, but can be related to flexible nanostructures with a lower elastic modulus than the larger solid material [28].

Forces that are applied to an area of nanostructures in Figure 6A–D may not be enough to heavily deform nanostructures; however, the force may be enough to move nanostructures laterally throughout contact, lowering the resolution and measured height. Self-imaging may be one effect that may occur due to the complexity of the nanostructured surface. Tip convolution is also a common over surface features. A large spherical tip traces a lower resolution path of the narrower nanostructure surface compared to the small and sharper conical tips. As the tip moves over the edge of a nanostructure, imaging can be switched to the tip, where tip convolution is shown as a representation of the tip. Figure 6A–D show the tip convolution of a large spherical tip over high aspect ratio nanostructures imaged as spherical nanostructures. Larger spacings between the nanostructures sometimes show the switching back to measuring the surface. Switching can also relate to impressions and effects of closely related structures at different heights, with higher features sometimes covering or reducing other features. Partial spheres are consistently visible in Figure 6A–D. The large spherical tip begins scanning these nanostructures imaged as spheres until a point is reached during the scan where the tip encounters another nanostructure of near or increased height, which moves the tip out of contact with the original nanostructure. Many of these partial sphere images shown in the scans are high nanostructures. These are represented as brighter structures from the z-axis colorized scale bar, where brighter means higher and darker means lower in height. Contact being lost for a nanostructure being considerably higher than the surrounding ones would indicate that the loss of contact is due to movement of the nanostructure either laterally away from the cantilever path, as the tip slips off and moves to the next nanostructure/s contact, or in the negative z-axis due to collapse. The observation suggests that currently, only one nanostructure is probably in contact with the tip, causing the observed movement or deformation [29,30,31].

To investigate the partial sphere anomaly further, the slow scan direction (y-axis) was measured beyond the clear cut-off point where the nanostructure becomes invisible in the scan. Remarkably, this anomaly was predominantly observed in high individual nanostructures, and no other nanostructures appeared within a relative height range. Figure 7B shows that the partial sphere nanostructure was 38 nm higher than the next nanostructure down. Figure 7C,D show three colored arrows that are placed on notable nanostructure patterns for each scan as reference points for the partial spheres. Five shorter white arrows indicate partial spheres visible in Figure 7C; however, they have either disappeared or are shown to be highly reduced in the z-axis in Figure 7D. This indicates that the tip is not moving to scan another nanostructure in the local region, rather the individual nanostructure is being heavily damaged and topographically altered. Some of the partial spheres in Figure 7C relate to much lower and smaller structures in Figure 7D, displaying that the tip with a SetPoint force of 420 nN is collapsing and damaging these nanostructures, leading to a reasonable change in the nanostructure surface area, as visible by the increased percentage of black area in the middle to lower scan area in Figure 7D. Despite minor drift between the two scans, roughness measurements were compared in Figure 7C,D cropped to the same size, as shown in the table in Figure 7E. There is a slight drop in the roughness values in the two scans. A reduction in the overall number of nanostructures and the higher nanostructures in the scans indicate high damage of a moderate percentage to the nanostructures in this small area. Figure 7F,G display an increase in friction over the partial sphere area at the position shown in Figure 7B. The increase in friction coefficient and force, shown by the red arrows, is at the point where the tip moves off the partial sphere and onto the other lower nanostructure. This marks the juncture between the breakdown or removal of the higher nanostructures and the subsequent reconnection to a more stable nanostructure. The tip is likely to contact several nanostructures at this stage in the surrounding region, as there are several local nanostructures at around the same z-axis color in the scale bar.

### 3.6. High Setpoint Friction Analysis

Measurements were performed across the very high SetPoint force scan of 5200 nN, which shows notable damage to the nanostructure surface. Figure 8A,B show plots across the edge of the Figure 6I scan, indicating the surface change in height in the damaged nanostructures. A 23 ± 8 nm reduction was measured across one region. Overall heights of these nanostructures average between 500 and 700 nm from the base. This equates to a 6.5% reduction in the overall nanostructure length. Although this is a small reduction statistically, the damage substantially changed the surface geometry, as seen in Figure 8A. The average friction force for a high spring constant spherical cantilever on the same sample at a SetPoint force of 320 nN is 6 nN, while the average friction force with a 5200 nN SetPoint force setting is approximately 1724 nN, as shown in Figure 8C,D. Values were significantly different between these two friction coefficient values for trace and retrace. These were mostly due to the opposing forces on the cantilever from the collapse and distortion of the nanostructures. Although the subsequent scan with a SetPoint force value of 520 nN flattens the nanostructure area slightly more, it is noteworthy that the nanostructures have already experienced significant collapse. Hence, the recorded friction coefficient aligns closely with the friction scan values depicted in Figure 6A–D, given that the scan predominantly assesses surface friction rather than the counteracting forces arising from the collapsing nanostructures.

### 3.7. Roughness Comparisons

Roughness scans were performed over the Figure 6 topography scans. Figure 6A–D show slight variances in RMS and Ra roughness as the scan size increases, with the differences being within a 6 nm variation. Notably, the scan with a high SetPoint force of 5200 nN measured a lower roughness than Figure 6A–D from the flattening of the nanostructures. Lowing the SetPoint force to 520 nN and scanning over a 1 × 1 µm larger area saw the roughness increase due to the unflattened nanostructures on the scan edges. Cropping the Figure 6F 520 nN scan down to a 9 × 9 µm size, the same area as in Figure 6E saw higher nanostructures on the edge of the scan. This is due to the curve of the spherical cantilever not compressing the edge nanostructures as far. The cropped scan area sees a small rise in roughness, partially due to some of the less damaged nanostructures on the scan edge and partially due to a level of elastic recovery of the nanostructures, as shown in Figure 9. This plots the difference in heights measured at 5.5 µm horizontally across the middle of the scan. Recovery of around 20 nm displays a mostly plastic deformation with minor elastic recovery, as shown in the Figure 6G scan and the Figure 10F plot. The surface does not restore to a recognizable nanostructure pattern, as visible in Figure 10F. The damage by these forces destroys the original nanostructure patterning. Figure 10A–D sees a gradual increase in nanostructure heights as the increased force pushes harder into the nanostructure surface, overcoming short-range force and atmospheric water layer barriers as well as contacting slightly lower nanostructures. These nanostructure peak patterns shown in Figure 10A–D are lost on too high of a force, as shown in Figure 10E,F [32].

The friction coefficient and friction force values can be acquired from contact mode scans by the accompanying lateral trace and retracing the scans gathered simultaneously. Figure 11 presents the average friction force values for scans at contact mode SetPoint forces of 320, 370, 420, and 470 nN over the AMK surface. Averaged friction forces, calculated in Excel, were lower than expected. The 320 nN SetPoint friction force is around 6 nN, and with SetPoint force jumps of 50 nN, subsequent values were 10, 14, and 16 nN. These jumps in friction force with an evenly increasing SetPoint value reduce and begin to taper off, appearing to approach a maximum value. Damage from very high forces may occur from the handling or any high-level contact force (≥µN) of these nanostructured surfaces. These surfaces are primarily designed for anti-bacterial implants, and high-force contact is most likely to occur between manufacturing and fixation inside of the body. Implant surface contact with neighboring surfaces in this period will generally be higher than the nano-Newton forces applied by experiments in this report. In shipping, handling, and especially heavy contact by orthopedic surgeons to position the implant inside the body, forces would be expected to be several tens of Newtons. Roughly mimicking this interaction to the status of how the nanostructure array will recover and the change in properties that high forces will cause to the functionality, a 5200 nN (5.2 µN) force was applied to an area of the nanostructures. This emphasizes the vulnerability and susceptibility to deformation of these nanostructure surfaces. It is crucial to take this into account as a potential surface condition during the period between production and implantation. While the collapsed nanostructured surface exhibited elevated roughness compared to a non-nanostructured surface, its z-axis roughness was approximately 50% of the roughness measured in the original undamaged nanostructures. Collapsed nanostructures will be very random in patterning; however, when they collapse, there a non-uniform surface with a reasonable level of friction and roughness that is expected to exhibit a level of anti-bacterial behavior above the smoother and flatter control surface will remain.

### 3.8. Directional Friction across the Nanostructures

The spring constant of a cantilever with a 5 µm spherical tip was measured at 36.2 N/m by performing a force curve on a glass microscope slide. Contact mode scanning measured lateral deflection trace (black) and retrace (red) tip response over a high nanostructure. Figure 12A,B show a slightly larger nanostructure compared to those surrounding it, measuring over 1 µm in diameter. Tip convolution explains the width of these AMK nanostructures as they average below 100 nm in diameter and were verified by SEM [2]. A dip in the middle region of the nanostructure is an area of lower friction surrounded by step-edge effects. The smooth trace scan, shown in Figure 12B, differs from the very noisy retrace plot (in red) overlapping the nanostructure in a similar pattern. Retrace noise is supported by a plot over the smaller and shorter nanostructure in Figure 12C,D measured with a higher SetPoint force of 100 nN compared to 20 nN in Figure 12A,B, showing again a smooth trace scan and an even noisier retrace scan. There is a phenomenon that can be counted toward this effect, such as damage to the nanostructure or the tip on the first pass, which has not affected the side walls of the nanostructure. The SetPoint forces are quite low, and the trace and retrace plots map each other with only minor errors to discount any real damage effects. The most likely cause of this effect is directional scanning. Directional tilting of the nanostructures, nanostructure roughness, and torsional flexion preference of the cantilever can contribute heavily to this patterning. Specifically, the tilt angle of the cantilever with respect to the nanostructure pattern can significantly impact the mapping process [33,34,35]. This asymmetric directional patterning of nanostructure lateral deflection may involve directional roughness present by the wear or damage from the lateral deflection trace scan; directional roughness and the directional patterning of nanostructures have an impact not only on the noise generated from the contact between rough surfaces but also on the way liquids interact with these surfaces. The hydrophobic AR-Ti surface changes to super hydrophilic when etched to an AMK nanostructured surface. This change results in Wenzel’s behavior of the atmospheric water layer [3]. Liquids tend to move more rapidly in the direction of the surface angles, with droplet velocity potentially reaching speeds up to six times faster than when moving against the direction of the nanostructures. This will affect liquid resistance as the tip moves through the atmospheric water layer, as shown below in the case of a single droplet with the receding edge pinned to the nanostructure surface, as shown in Figure 2.

Figure 12 illustrates lateral deflection trace and retrace plots showing minimal variations in relation to each other. Frequently, the lateral deflection trace and retrace exhibit height differences corresponding to directional variations in friction values. Figure 13 exemplifies this phenomenon over a single nanostructure. The trace and retrace differences are approximately 0.02 V, with a calculated friction force and a SetPoint force of 200 nN equaling 4 nN of friction force. The retrace (red) again displays noisier behavior on the horizontal plane areas of the plot and smoother behavior on the side walls. The difference is shown in the trace–retrace scan contrast in Figure 13B. The trace scan images in Figure 12 and Figure 13 appear smoother than the higher detail present in the retrace scan images. Along with this higher detail appearing in the scanned image, higher frictional roughness or noise also appears. In the 2D plot, the trace scan, despite the noise, appears sharper and more detailed. The two peaks in the trace plot show more definition and sharper contrast, despite the smoothing and blurriness of the trace scan image compared to the retrace.

### 3.9. Individual AMK Nanostructure Friction and Roughness

Contact mode scans shown in Figure 14 below display improved scan definition due to changing the cantilever from a spherical tip to a conical tip with a tip apex of <10 nm. Lateral Force Microscopy deflection trace and retrace scan images at 2 × 2 µm highlight an individual AMK nanostructure, showing a high contrast in the 2D plot in Figure 15B. Elevated features on the top of the nanostructure measure closely matching friction levels, as the trace (black) and retrace (red) friction coefficient in the 2D plot almost overlap each other, while the flatter areas of the nanostructure peak map have a trace–retrace difference of 0.011 V, which relates to a friction force of 0.25 nN. This is only a small change in friction. However, the contrast is most likely due to the relative uniformity of the motion of the cantilever over the elevated features, while the flat regions are at a significant angle to the horizontal plane and are visible in the 3D representation shown in Figure 14D. RMS roughness over one of these nanostructure peaks within the blue square in Figure 14C is 5.9 nm, which is considerably rough compared to the low friction force.

In Figure 15, a closer view of a 1 × 1 µm area highlights these raised features, which range in height from 9 to 26 nm. The average friction coefficient was determined to be 0.006 V, corresponding to a friction force of 0.13 nN, with a SetPoint value of 37.7 nN. These elevated features are of the same material as the rest of the AMK nanostructures and are expected to be a result of the nanostructure growth by hydrothermal etching. These smaller protruding rectangular features measure tens of nanometers in width and up to 350 nm in length. The advantage of these smaller structures toward anti-bacterial and osteointegration functionality is an increase in roughness of the nanostructure surface, which improves osteointegration and possibly increases stress to interacting bacteria. A rough surface also reduces bacteria macromolecule tethering, lowering overall bacterial adhesion points on the surface [36].

## 4. Conclusions

Performing tapping mode topography over Ti6Al4V anti-bacterial nanostructured surfaces resulted in measurable damage. Tapping over nanostructures provided enough force to indent areas of nanostructures to a considerable depth in relation to the top nanostructure plane. Considerable damage and deformation were recorded from both tapping mode and contact mode topography scanning. Indentation at tens of nanometers was mostly plastic, as subsequent scans recorded minimal recovery up to an hour later. There was a small amount of elastic recovery; however, permanent damage resulted and would alter the functionality of the surface. A drop in roughness would change the interaction behavior of these nanostructures purposed for anti-bacterial functionality, as well as a measured lowering of the number of viable nanostructures per unit of surface area. An expected reduction in bactericidal behavior can result from improper handling and heavy contact forces on these potential implant surfaces. This report highlights the limitations of Ti6Al4V nanostructured surfaces, mainly toward permanent damage during an applied high nN to low µN range force.

The partial sphere effect presented in scan images of Ti6Al4V nanostructured surfaces with applied forces at hundreds of nano-Newtons is often related to the tip connecting with another area of the surface, such as another nanostructure, during scanning. While this is a viable effect to occur over a nanostructured surface with different length nanostructures, it is not the only explanation. This effect can also occur in the collapse, removal, or heavy and immediate damage to the nanostructure surface. This explanation was verified by a follow-up scan of the area showing the collapsed, heavily damaged, or removed nanostructures. Contact mode imaging was also performed as a continued impact and force overscan areas rather than intermediate impact from tapping mode. Again, mostly plastic damage was resultant of the scans. After sustaining damage, the nanostructured surface did not fully recover its initial pattern. However, it preserved a moderately rough and irregular surface morphology, which could still be advantageous for applications, such as anti-bacterial implant surfaces. In essence, the experimental investigation and analysis of damage behavior in these tall and thin nanostructures revealed their limitations and potential functional uses. This research emphasizes the significance of thorough testing for novel biomaterials that incorporate similar nanostructured surfaces, guaranteeing their appropriateness for intended purposes.

## Data Availability

Data can be made available upon request from the authors.

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
