# Peer review of "Damage Behavior with Atomic Force Microscopy on Anti-Bacterial Nanostructure Arrays"

_nanomaterials, 2024, doi:10.3390/nano14030253_

Round 1
Reviewer 1 Report
Comments and Suggestions for Authors
In this paper, the authors comment on atomic force microscopy (AFM) experiments they have performed on a Ti alloy nanostructured surface. Even in the tapping mode, they observe permanent surface damage produced by topographical scan. They compare the mechanical effects of a conical SiN tip and a spherical SiO2 tip with much larger curvature radius at the apex. In routine experiments (normal scan conditions for topographical characterization), both tips can produce plastic deformation of the surface nanostructures. This deformation weakens the antibacterial function of the surface, a non-desirable effect for medical implants. According to the authors, macroscopic handling tools and even the surgeon fingers could produce large-area modifications of the nanostructured surface.
The paper is well written, the methodology used is good and the experiments are well described. Although not original, interesting image anomalies are discussed, such as edge effects, self-imaging of a large tip by much narrower surface nanostructures, and convolution of the topography by the tip shape. Given the wide use of AFM as surface characterization tool, the paper is of real interest and deserves a publication in nanomaterials.
Before, a few minor points should be addressed by the authors.
1) Unless this reviewer missed the point, the scan direction is not specified. It is written that the scan started from the lower left corner. The question is: upward or to the right? In direct connection with that question, one observes horizontal bands with different gray contrasts in Figs. 3A, 3B, 3D, 5B. What is their origin?
2) Fig. 5, please mark the line along which the 2D plot (Fig. 5C) was extracted.
3) Page 9, the authors mention the effects of the atmospheric water layer on the AFM experiment. Can they say something about a possible hydrophobicity of the nanostructures surface compared to the as-received Ti alloy sample?
4) Page 9, line 316, the authors write that the setting point force started at 370 nN for contact mode experiments. The same force is indicated as 320 nN in line 335 of the paper as well as in the caption of Fig. 7. Please check.
5) A general question related to the discussion in the second half of Page 15 is the following. Can we really extrapolate the nanostructure collapse produced by a microscopic tip (apex radius of a few µm) loaded by µN force to the macroscopic world (10 N force exerted by the surgeon on a few mm² nanostructured surface of an implant)? The average pressure is of the same order of magnitude in both cases, but is it enough to draw a definitive conclusion?
6) What would be the recommendations of the authors regarding the tip shape and size, cantilever force constant ... for AFM topographical analysis of a nanostructured surface of a metallic sample in order to minimize the damage?
Reviewer 2 Report
Comments and Suggestions for Authors
The authors studied about surface properties and damage behavior with atomic force microscopy on high-aspect ratio nanostructure (Ti6Al4V) arrays with application of rough and non-uniform morphology for an anti-bacterial implant surface. There are some issues as follows.
1. The purpose of this work is clearly described in introduction section.
2. Title is not clearly described on the purpose of this work.
3. In the abstract, the authors claimed application of rough and non-uniform morphology for an anti-bacterial implant surface. However there is no experimental evidences for this.
4. Define AMK
5. The authors should argue tip wear issue.
6. The model shape of the nanostructured surface (schematic 1) is not matched on the SEM image of Fig. 1(b)
7. AFM image quality is not good enough.
8. The nanostructure has not the shape of high-aspect structure.
Comments on the Quality of English LanguageIt is difficult to understand on some points.
Reviewer 3 Report
Comments and Suggestions for Authors
The presented results on the behavior of the surface of hydrothermally etched nanostructures of 32 Ti6Al4V under mechanical load by the AFM tip are quite interesting. However, before publication, some issues have to be addressed.
The AFM gives information from a relatively small region of the sample. Thus, the results obtained from one region can significantly differ from the other. Thus, I would not trust data (such as RMS) obtained and calculated only from one region. A proper statistic is absolutely required. Data and explanations from a limited number of scans are insufficient to derive reliable conclusions.
I mean no harm in any way, but it would be beneficial to the quality of the manuscript if the authors would contact an expert in the AFM field who could suggest a better approach for designing the experiments and for data analysis.
As a general remark, I suggest that the authors increase the images' quality. This is especially important for the AFM images; scales (x-y and z) are often difficult or impossible to read. The size of the font should be increased almost in all cases.
What is more, I have some specific comments for the manuscript.
1. line 58 - The surface roughness (Ra) parameter is a strong function of the scanning area. Thus, specifying the scanning area for the Ra parameter would be beneficial.
2. The NT-MDT cantilevers data is inconsistent with the available information; for example, the NSG30 spring constant is given in the 20 – 70 N/m range, while the information on the website states the 22 – 100 N/m range. According to the producer, they are N-doped silicon, not positive-doped.
3. There is no scale in the Figure 1 A.
4. Line 121 – the ‘spacing’ parameter is not defined.
5. Line 164 – the amplitude and spring constant do not give information about the SetPoint force; thus, I recommend removing ‘force’ from the description, leaving just the ‘setpoint.’ I would correct this throughout the whole manuscript.
6. Line 183 – the authors keep mentioning tip contact force very vaguely. Some estimation of the actual tip contact force would be very useful.
7. In Figures 3, 4, and 5, artifacts connected with the scanning process are not removed; aligning or leveling the rows using Gwyddion software would be beneficial.
8. Line 199 – the values of the RMS drop are given for a particular case (1 image) – I would recommend significantly increasing the number of measurements or points from where the calculations of the RMS are performed.
9. Figure 5 – there is no indication of the line along which the cross-section is made. Moreover, the scale and the height values on the B are impossible to read.
10. There is no Figure 6. Instead, there is Schematic 1. I would rename it to Figure 6 for consistency.
11. Line 315 – the 11-degree angle cannot be ‘added’ to the force. Force is a force; angle is an angle. The setpoint is measured from the deflection of the cantilever.
12. The data presented in Table 2 is too accidental; such comparisons should be performed for several cases in different sample regions. Analysis from one region of the sample is prone to producing unreliable results.
13. The discussion in subsection 3.8 is based on minimal differences in the lateral force trace and retrace images. Without more examples, they may not originate from the sample but are simply the effect of the tip geometry.
14. I need clarification with the addition of Figure 16. It shows a different morphology than the initial images, especially comparing it to Figure 2. This looks like a completely different sample.
To summarize, the manuscript topic is quite interesting. However, its quality needs to be improved by more careful analysis and better presentation.
Round 2
Reviewer 2 Report
Comments and Suggestions for Authors
The authors intensitvely revised the manuscript. But, still the image quality if not enough for publication. Especailly, the size of all numbers of the figure is not small, thus it's difficult to recognize. And the height information of all AFM figures are not corrected to the zero nm for offset calibration.
Comments on the Quality of English LanguageBetter to be checked.
Reviewer 3 Report
Comments and Suggestions for Authors
The authors made numerous improvements to the manuscript. However, some issues have to be addressed before publication. I still have some doubts about the reproducibility of the measurements. Still, many presented results are based only on one AFM image. I would try to perform extensive statistics and show specific images only as an example of the effect.
Moreover, I have some specific comments.
1) The authors should reference the Gwyddion software, which can be found on the Gwyddion webpage.
2) Line 113 – missing a dot at the end of a sentence.
3) The artifact connected with the scanning process visible in Figure 4 A has still not been removed.
4) I did not suggest this before, but the article would benefit from force-distance curve measurements. Especially using a spherical tip to avoid indentation. Then, the authors could see the reaction of the nanostructures from the tip and investigate elastic and plastic deformation more directly.
5) The scales visible in Figure 8 C, D, and E images should be expressed in µm, or the number of decimal places should be limited. For example, 2.00E-06 m should be simply 2E-6, or 2 [µm]. The x-axis description in image D should be moved. The y-axis description in D and E should be changed to explain what physical properties it describes; Volts (in Volts) are just units.
6) The data in Table 2 should be explained better; the number in brackets in the surface area is not explained in the table caption. Also, the Ra parameter is well explained in the Gwyddion documentation, but for the general reader, it would be helpful to show the difference between RMS and Ra parameters.
7) I suggest removing Figure 11 if the average values are as specified; there is no point in showing straight lines with the same values.
I can’t stress enough that the authors should really focus on a good presentation of the AFM data. Especially when the article is almost solely based on AFM images. As a general remark, it would be beneficial if the scale bar presented in various images had its description. For example, in Figure 1, the scale should have 500 nm written above it. I know there is information in the caption, but having it on the image is much easier to read.
Moreover, the quality of the images, especially the scales, is still insufficient. Authors need to increase the font size significantly. Also, please choose either a scale bar, preferably with the value, or a scale on the side of the image. The Z-scale should also be clearly visible, which is not the case at the moment. There is also an option in the Gwyddion software to export higher-resolution images.
